# Electrochemical Characterization of Diffusion in Polymeric vs. Monomeric Solvents

**DOI:** 10.3390/ijms25084472

**Published:** 2024-04-18

**Authors:** Ze’ev Porat

**Affiliations:** 1Department of Civil and Environmental Engineering, Ben-Gurion University of the Negev, Be’er-Shava 8410501, Israel; poratze@post.bgu.ac.il; 2Department of Chemistry, Nuclear Research Center-Negev, P.O. Box 9001, Be’er-Sheva 8419000, Israel

**Keywords:** polymer electrolytes, microband electrodes, linear diffusion, Fick’s equations, diffusion in polymers

## Abstract

Polymer electrolyte was used as a medium for testing the performance of microband electrodes under conditions of linear diffusion. Cyclic voltammetry (CV) and chronoamperometry (CA) experiments were performed in a highly viscous medium, where diffusion rates are much slower than in fluid solutions. The log i vs. log v (CV) or log i vs. log t (CA) relationships with the current equation confirmed the existence of such conditions, yielding slope values that were lower than the expected 0.5. This could indicate an impure linear diffusion profile, i.e., some contribution from radial diffusion (edge effects). However, the desired value of 0.5 was obtained when performing these tests in monomeric solvents of similar viscosities, such as glycerol or propylene glycol. These results led to the conclusion that the current equations, which are based on Fick’s laws, may not be applicable for polymer electrolytes, where various obstructions to free diffusion result in a more complicated process than for monomeric solvents.

## 1. Introduction

Polymer electrolytes have been used in the past decades as media for electrochemical processes. In particular, polymers bearing polyether chains are able to dissolve considerable amounts of electrolytes such as lithium salts, which provide electrical conductivity to the system [1,2,3,4,5,6]. Nevertheless, the conductivity in polymer electrolytes is still lower than in conventional electrolytic solutions; therefore, electrochemical experiments conducted in such media must employ microelectrodes, which are less sensitive to solution resistance than regular-sized electrodes [7]. However, a problem can arise from the use of microelectrodes, such as microdisks or microspheres, when exceedingly low currents are present, especially in very viscous solutions where the diffusion rates are very slow. A practical way to overcome this difficulty is using microband electrodes. These electrodes have widths on the micrometric scale; this maintains the properties of microelectrodes while their lengths are on the millimetric scale, providing considerably higher currents [8]. Microband electrodes can be fabricated by either photolithographic techniques, or by sealing thin metallic film or foil between two insulators [9]. In the past, we reported the fabrication of microband electrodes via sealing procedures, and tested the microband electrodes [10] to demonstrate their performance in viscous and resistive polymeric solutions [11].

In the course of characterizing microband electrodes, cyclic voltammetry (CV) experiments have been performed under conditions of radial and linear diffusion (Figure 1). Achieving linear diffusion profiles with microelectrodes on the micrometric scale in cyclic voltammetry measurements requires either fast potential scan rates [12] or significant suppression of the diffusion rate. The latter be accomplished by controlling experimental parameters such as the size and the shape of the diffusing probe, the temperature and the viscosity of the solution [13]. As compared to conventional fluid solvents, electrochemical measurements in polymer electrolytes have shown diffusion coefficients that are reduced by several orders of magnitude [14,15,16]. As one of the means to obtain pure linear diffusion with negligible contribution from edge effects, it was an obvious choice to conduct these tests in polymeric solutions.

An estimation of the diffusion geometry can be obtained using the dimensionless time parameter *τ*, defined as follows:*τ* = 4*Dt*/*r*^2^(1)
where *r* is the electrode’s radius, *D* is the diffusion coefficient and *t* is the electrolysis characteristic time. For cyclic voltammetry, *τ* equals to *RT/nFv,* where *v* is the potential scan rate. When *τ* = 1, the diffusion distance of a solute is equal to the radius of the electrode. When *τ* >> 1, the thickness of the diffusion layer *δ* is greater than *r* and the diffusion geometry is radial; however, when *τ* << 1, the diffusion layer is thin relative to *r*, and thus the diffusion profile is linear [7].

The approximate thickness of the diffusion layer at the electrode’s surface is given by the following [17]:*δ* = (2*Dt*)^1/2^(2)

If we define a limiting thickness that does not exceed one tenth of the electrode’s smallest dimension (i.e., *δ* = 0.1*r*), within which the diffusion geometry is purely linear, we obtain the following (after substituting the expression for *t*):0.1*r* < (2*DRT*/*nFv*)1/2(3)

Taking *D* = 10^−8^ cm^2^/s and *r* = 3 μm (the width of our microband electrode) at ambient temperature, linear diffusion should be predominant at scan rates greater than 0.7 V/s. Substituting this value into Equation (1) yields *τ* = 0.02, which is considerably smaller than 1, as should it be for linear diffusion. At lower temperatures, *τ* would be even smaller.

The experimental criterion for the existence of linear diffusion conditions can be derived from Equation (4) for the peak current in cyclic voltammetry [17], as follows: *i_p_* = 0.4463*nFAC*_0_(*nF/RT*)^½^ *D*^½^ *v*^½^(4)
where *A* is the electrode’s area and *C*_0_ is the concentration of the electroactive species. According to this equation, *i_p_* is a linear function of *v*^½^. In the case of linear diffusion, a straight line with a slope of 0.5 should result after conducting a series of CV experiments at a succession of potential scan rates and plotting the results in the form of log *i_p_* vs. log *v*.

In this paper, we present the results of such tests in polypropylene glycol (PPG). Despite the expectations for pure linear diffusion, based on the above-mentioned calculation, slopes not higher than 0.4 were observed under all experimental conditions. This was investigated and compared with results obtained in monomeric solvents of similarly high viscosities, and the different modes of diffusion in these systems are discussed.

## 2. Results

The performance of the microband electrodes, which were manufactured in our laboratory, was tested in a solution of polypropylene glycol with an average molecular weight of 4000 (PPG 4000). The addition of lithium salt (LiSO_3_CF_3_, 1.1 M) as an electrolyte renders the polymer melt very viscous, due to the cross-linking of the polyether chains via coordinative bonds with the lithium ions. If pure linear diffusion profile is required at the very narrow microband, high viscosity is necessary for the suppression of the diffusion coefficient of the electroactive molecules. The higher the diffusion coefficient, the faster is the development of the diffusion layer in the vicinity of the electrode, increasing contributions from unwanted radial diffusion.

The linearity of the diffusion profile was deduced from a plot of log i_p_ vs. log v for a series of cyclic voltammograms at various scan rates. Since Equation (4) is valid only under conditions of linear diffusion, a slope of 0.5 would be indicative for the existence of such conditions. Examples for two of the voltammograms, recorded at room temperature at 1 and 200 mV/s, are shown in Figure 2.

Curve A was obtained at a low scan rate and represents a mixed diffusion profile, while curve B was recorded at a faster scan rate and shows a higher contribution of linear diffusion. The plot of log i vs. log v is presented in Figure 3. Line A is drawn for a set of measurements carried out at room temperature; the slope of 0.4 may indicate that the diffusion profile is mainly linear, with some contribution from radial diffusion. We considered the possibly radial contribution to be a result of insufficient viscosity that may have arisen from either insufficiently low temperature or from residual amounts of ethanol. The ethanol was used to facilitate the dissolution of the ferrocene probe in the polymer, and was later removed by evaporation. That residual ethanol may have acted as a plasticizer, and thus increased the diffusion rate in the polymer. These two possibilities were checked experimentally, as described in the following.

Further removal of possible residues of ethanol was carried out via vacuum pumping under more drastic conditions: The polymeric solution was heated at a higher temperature (90 °C) during the evaporation, in order to increase its fluidity and enable any traces of ethanol to diffuse out more easily. Furthermore, the duration of the evaporation was longer (three days), and the vessel was held in the oil bath in a tilted position so that the surface area of the polymeric solution was larger and its depth was smaller. Finally, some of the solutions were spread as a layer on the surface of the electrode assembly rod and vacuum pumped for three days. Nevertheless, no significant change in the value of the slope was observed.

The effect of temperature was tested by performing two more sets of measurements at 12.6 and 2.6 °C, and their analyses are presented in Figure 3 (lines B and C, respectively). Surprisingly, the trend in these results was in the opposite direction than expected: the slope values decreased with decreasing temperature, rather than approach the value of 0.5. A possible explanation for this discrepancy could be the increased uncompensated solution resistance (*iR_u_*) at lower temperatures, which causes flattening of the peaks, and thus lower measured currents. This effect was first tested using digital simulations of CV experiments, which compared two systems that were different from each other by the magnitude of *R_u_* only. It was found that decreasing the magnitude of *R_u_* from 500 MΩ to 100 MΩ results in an increase in the log i–log v slope from 0.40 to 0.46, in agreement with the assumption above. Experimental verification of this effect was carried out by taking three different steps: (a) Decreasing the solution resistance by raising the electrolyte concentration; (b) decreasing the *iR* product using lower concentration of the electroactive species to obtain lower currents; and (c) applying positive feedback for *iR* compensation. None of these steps seemed to cause a meaningful change in the magnitudes of the slopes, which stayed within the range of 0.34–0.43. In testing steps (a) and (b), a similar trend in the slopes was observed with decreasing the temperature, although to a lesser extent. Testing the effect of positive feedback in two solutions with different electrolyte concentrations, 1.1 M and 0.7 M, yielded no change in the slope in either case, as shown in Figure 4. This is in spite of the remarkable reduction in the peak-to-peak separation (ΔEp), which indicates a smaller *iR* drop. Thus, it was concluded that solution resistance is not the factor responsible for the decreasing slopes at lower temperatures, or for not achieving the target slope of 0.5. 

Another means of increasing the solution viscosity was using higher molecular weight polypropylene glycol as the solvent, e.g., PPG 10,000. Upon dissolving the same concentration of electrolyte (O/Li = 16), a very viscous melt was obtained, in which well-shaped voltammograms were recorded in the scan rate range of 1–500 mV/s (Figure 5A). Plotting the data as log i vs. log v (Figure 5B) revealed another unexpected result: a slope of 0.26, which is considerably lower than the slopes obtained with PPG 4000. Repeating the experiment in this solution, with a wider microband (w = 4.6 μm) and with two microdisk electrodes of 25 and 50 μm radii, yielded higher slope values of 0.35, 0.38 and 0.40, respectively. These results seem to indicate, on one hand, the existence of some radial contribution to the diffusion profile, since these slopes approached 0.5 as the ratios between the perimeters of the electrodes to their surface areas became smaller, i.e., the electrodes are less sensitive to edge effects. On the other hand, even with the 50-micrometer-radius microdisk electrode, a value not higher than 0.4 was obtained; this points to the possibility that such slopes cannot be obtained in viscous polymeric solvents at all.

The rationale of this conclusion may be found in the basic difference between the diffusion profiles in conventional monomeric solvents such as water, alcohols, nitriles, etc., and polymeric solvents. In monomeric solvents, free diffusion of the solute occurs with actually no physical restriction. Polymeric solvents, however, can be entangled, cross-linked or contain local aggregates or orderings, which can all introduce constraints to the diffusion paths. Therefore, equations derived from Fick’s laws, which were developed for conditions of free diffusion (i.e., in monomeric solvents), may not be applicable in polymeric solvents, especially where such restrictions exist. In the case presented here, the reason for not obtaining slopes of 0.5 is that the current equation (Equation (4)) cannot be used in its present form, even though it is clear that the diffusion profile is nearly linear. One of the changes in the current equation should be in the function of *v*; its power in such cases should be smaller than 0.5.

Experimental verification of this idea was conducted in propylene glycol and glycerol, two monomeric solvents at low temperatures that are rather viscous, even at room temperature. In propylene glycol, voltammograms at various scan rates were recorded at three different temperatures, using TTF-(OCH_2_)_3_CH_3_ as the electroactive probe. A representative voltammogram, recorded at −23 °C, is shown in Figure 6 (inset); it has two well-resolved couples, and the currents of the first oxidation peaks were used for the plot of log i vs. log v, which is shown in Figure 6. At −23.7 °C, a well-defined straight line was obtained, which exhibits the desired slope of exactly 0.5. This is actually the first time that pure linear diffusion has been observed with a microband electrode, and, perhaps more significantly, this result is in accordance with the new concept of different diffusion rules in the two kinds of solvents. At higher temperatures, −12 °C and 1 °C, a slope of 0.5 was obtained at faster scan rates, which were required in order to compete with the growing contribution from radial diffusion.

With glycerol as the medium, the composition of the solution was similar to that of the PPG 4000 solution in which the first experiments were performed (10 mM FcCO_2_PEG(350)CH_3_, 1 M LiClO_4_), with the exception of the solvent. The measurements were carried out at −27 °C, at scan rates ranging from 1 to 200 mV/s, all yielding peak currents that aligned on a straight line with a slope of 0.49 (Figure 7). Under these conditions, it was possible to obtain pure linear diffusion at a 2.7-micrometer-wide microband at a scan rate as low as 1 mV/s (Figure 7 inset).

In comparison with the results obtained in propylene glycol, a solution with the same probe in polypropylene glycol was prepared, and voltammograms were recorded at room temperatures, keeping all other experimental parameters unchanged. A typical voltammogram is shown in Figure 8 inset, having two main features: (a) The first oxidation peak is less resolved than in propylene glycol, but is still distinguishable; (b) the other peaks have sharp and symmetrical shapes, which are typical to surface-immobilized systems such as monolayers or thin-layer cells. This indicates that these bulky probe molecules are trapped in the polymer, and their freedom to diffuse is so limited that only molecules in the near vicinity of the electrode can undergo a redox process within the time scale of the experiments. A plot of log i vs. log v for the first oxidation peak (that appear as a shoulder in the voltammogram) is linear with a slope of 0.35, which is close to the values that were previously measured in PPG 4000. More interestingly, such a plot for the first reduction peaks has a typical shape of CV in a thin-layer cell, and has a slope of 0.32 (Figure 8); thus, it should show a linear dependence of *i* on *v*. This apparent paradox may be regarded as a situation of quasi-immobilization, i.e., the molecules are not covalently bound to the electrode surface or trapped in a thin layer, but in principle can undergo diffusion, although in a restricted manner.

The unusual diffusional behavior in polymeric solvents was also tested using an independent electrochemical technique—chronoamperometry. Since the solution was dried under vacuum and the cell was kept tightly closed, the background curve could not be recorded in the same potential range prior to the dissolution of the substrate. Instead, it was recorded by stepping the potential from the start point to a value where no reaction occurs (point A on the inset CV, Figure 9) prior to recording the actual chronoamperogram at an equal potential step (point B). A typical response, obtained in the same solution as the CV experiment described in Figure 2, is shown in Figure 9. The corresponding Cottrell plot (Figure 9A) is based on the following Cottrell equation:*i* = *nFAD*^1/2^*Cπ*^−1/2^*t*^−1/2^(5)
where the result is linear, with zero intercept during the short time segment after the charging spike. At times longer than 2 s, a positive deviation is observed, which means that the rate of the current decay is slower than the Cottrell prediction. A similar deviation has been observed by Longmire et al. [7] in a system of ferrocene-labeled polyethylene glycol melt (which serves as both the electroactive moiety and electrolyte-dissolving solvent), and the extra current was explained by the enhanced flux of material to the electrode via radial diffusion. Regarding the unusual results of our CV experiments, we checked the log–log relationship between *i* and *t*, which, under conditions of regular diffusion, should be linear with a slope of −0.5. The plot, shown in Figure 10A, has three linear segments. Note that the first segment corresponds to a time interval of 1 s after the charging spike (2 s total) and has a slope of −0.49, in accordance with the Cottrell equation. This segment can be attributed to the solute molecules in the near vicinity of the electrode, which are capable of relatively undisturbed diffusion. The third segment corresponds to times longer than 3.6 s and has a slope of −0.32, which is close to the log i–log v slopes (0.33–0.40) obtained in the CV experiments that were performed in the same solutions. 

Chronoamperometry was also performed with the solution of PPG 10,000. The Cottrell plot (Figure 11) has a similar shape as that for the PPG 4000 solution, but the deviation from Cottrell behavior starts after 0.6 s. Here too, the plot of log i vs. log t includes three linear segments. The first segment corresponds to a total time interval of 0.6 s, and also has a slope of −0.49, while the third segment, corresponding to times longer than 2 s, has a lower value of −0.28. In this case, too, the magnitude of the latter slope is very close to that of the log i–log v slope (0.26) obtained in the CV experiment for the same solution.

## 3. Discussion

The attempts to find suitable experimental conditions to characterize microband electrodes under a linear diffusion profile led us to consider the complicated manner of diffusion in polymeric solvents, in contrast to free diffusion in monomeric solvents. In the latter, the three dimensional random-walk model holds, which the Fick’s laws are based upon. The expressions for the peak current in cyclic voltammetry (Equation (1)) and for the time dependence of the current in chronoamperometry (Equation (5), Cottrell equation) are both based on Fick’s laws; thus, they are valid only under conditions of Fickian diffusion. In polymer melts, however, a number of obstructions to diffusion exist, such as local entanglements, coordinative cross-links, blocking ions or ionic aggregates and short-range local ordering; therefore, these equations may not be applicable in all cases. In solvent-swollen polymers, wide voids or channels between polymer chains, nearly Fickian diffusion is possible, so such constraints may not be very significant. 

Masaro and Zhu, in their comprehensive review article on diffusion in polymers [18], indicated that this is indeed a complicated process, and its rate depends on the concentration and the swelling degree of the polymers. They distinguished between Fickian diffusion (case 1) and two types of non-Fickian diffusion: case 2 and anomalous diffusion. Fickian diffusion can be observed in polymers at temperatures considerably above the glass transition temperature (*T_g_*), or below *T_g_* but with the addition of a plasticizer such as water. In non-Fickian case 2, the solvent diffusion rate is faster than the relaxation of the polymer chains, whereas in anomalous diffusion, these rates are rather close. Fickian and case 2 diffusions are considered as the two limiting types of transport, and the diffusion distance *M_t_* is given by Equation (6) as follows: (6)$$M_{t}=kt^{n}$$

Another type of non-Fickian diffusion, super-case 2, was defined by Sperling [19]. These mechanisms were categorized based on the exponent *n*, as follows: Fickian: *n* < 1/2, anomalous: ½ < *n* < 1, case 2: *n* = 1 and super-case 2: *n* > 1. Grinsted et al. [20] also noted that water has a plasticizing effect on poly (methyl methacrylate) (PMMA). Therefore, diffusion of methanol in PMMA changes from case 2 to Fickian as the water content increases. 

Many studies and review articles were published on diffusion modes in polymers. Several diffusion models that are based on the obstruction effects were suggested, which consider the polymer chains to be stationary relative to the diffusing solvents or solutes, due to their much smaller self-diffusion coefficients [18]. Modeling of entangled polymer diffusion concerning entangled (reptational) homopolymer diffusion in melts and nanocomposites was extensively reviewed by Karatrantos et al. [21]. Diffusion in polymer electrolytes was widely discussed in a comprehensive review by Choo et al. [22]. However, they only referred to polymer–salt interactions at the segmental level and macroscopic ion transport based on the system of poly(ethylene oxide) and lithium bis(trifluoromethanesulfonyl)imide (PEO/LiTFSI). No additional solute was included in this system as the diffusing molecules, unlike in our case.

In our polymeric solution, the plasticizer was removed, so the polymer contained only the conductive electrolyte and the diffusing molecules. Due to the low temperatures and the high viscosity of the polymer electrolyte, we can assume that the relaxation of the polymer chains is very slow. In such unswollen polymers, the obstructions mentioned above may have a profound effect, especially when the diffusing molecules are large. Transport of the diffusant in such media will occur along paths of least resistance (PLR), which can be branched or have dead ends. This PLR network can reorganize with time as a result of polymer segmental motion, and create new paths for diffusion. In electrochemical measurements, an insufficient reorganization rate compared to the diffusivity of the electrophore leads to “fractal diffusion”. Thus, electrochemical current measurements would reflect a combination of the rate of reorganization, together with the rate of diffusion of the electrophore within the existing or changing PLR network. Therefore, the effect on the current equations is reflected not only in the exponent of the diffusion coefficient, but also in the time-related terms (*t* or *v*). This can explain why the exponents of *v* or *t* in all of our experiments are smaller than 0.5, which is the expected value for the Fickian profile of linear diffusion. The magnitudes of these exponents decrease as the experimental conditions increasingly impede fast reorganization, such as lower temperatures or higher molecular weight of the polymer.

This conclusion, which is based on empirical data, leads to generalization of the current equations for these voltammetric methods, in which the exponent of the time-related terms would be *n*/2, where *n* ≤ 1. The magnitude of *n* is a function of the nature of the diffusion environment, such as the reorganization time constant of the polymer electrolyte or the longer diffusion paths, and should reflect the properties of the system. These are affected by factors such as the temperature, the concentration and the valency of the electrolyte, as well as the structure of the polymer and its molecular weight. Their relative contributions yield complicated combinations that are specific to each system. Changes in the exponents of the time-related terms should imply additional changes in the current equations for CV and CA, in order to maintain the units of Amperes. Thus, the exponent of the diffusion coefficient, which is also a time-related term, should be changed accordingly. Thus, evaluation of the current equation for polymer electrolytes is needed, which should reflect the combined contributions of both processes to mass transport, i.e., polymer reorganization and solute diffusion. 

## 4. Materials and Methods

Equipment: A locally built low-current potentiostat was employed, and was controlled by a universal programmer (EG&G model PAR 175). All of the experiments were conducted inside a Faraday cage.

Materials: Ferrocene-labeled monomethyl poly(ethylene glycol) [FcCO_2_PEG(350)CH_3_] [23] and tetrathiafulvalene trimethoxy methyl [TTF-(OCH_2_)CH_3_] [24] were synthesized according to procedures described in the references. The polyethylene glycol 4000 (CAS No. 25322-68-3) and 10,000 (CAS No. 25322-68-3), glycerol (CAS No. 56-81-5) and propylene glycol (CAS No. 57-55-6) (Merck, Darmstadt, Germany) were of analytical grade, and were used without further purification. LiSO_3_CF_3_ (CAS No. 33454-82-9, Merck, analytical grade) was dried at 100 °C under vacuum and kept in a glove box.

Electrodes: The Pt microband electrode was 2.7 µm wide and 0.7 cm long, and was fabricated from commercial Pt microfoil [10]. It was separated by 2 µm thick Mylar films from two flanking 25 µm wide Ag microband pseudoreference electrodes, which themselves were flanked by the ends of two Pt rod counter electrodes (Figure 12). The Pt microfoil and flanking Ag foils and Pt rod counter electrodes were potted in an epoxy rod, the end of which was polished to expose the electrode assembly. This type of microband electrode with nearby Ag reference electrodes was fabricated expressly for use in semisolid, highly viscous polyether polymer electrolytes, in order to minimize uncompensated resistance effects. The combined electrode was inserted into the cell through the screw cap (Figure 13). Microdisk Pt electrodes of 25 and 50 µm radii with a nearby reference electrode were used. Their preparation is also described in Ref. [10]. 

Solutions preparation: The polymer electrolyte solutions were prepared by adding weighted amounts of the electroactive probe and the electrolyte to a weighted amount of the polymer melt in a Schott tube, which was equipped with a vacuum outlet and a tightly closed screw cap with a rubber/Teflon septum. Dissolution of the solids was facilitated by adding a small amount of ethanol or acetonitrile, which were later removed by heating at ca. 70 °C under vacuum. Solutions in monomeric solvents (glycerol and propylene glycol) were prepared without adding a co-solvent. 

## 5. Conclusions

Diffusion of molecules in polymer electrolyte melts is non-Fickian because of various perturbations to free random motion derived from concentration gradients.Therefore, the current equations for voltammetric measurements that were developed for fluid solutions are not valid in their regular form for such media.These equations are valid, albeit in very viscous solutions of monomeric solvents where the high viscosity is reflected by low diffusion coefficients.The modified current equations for voltammetric measurements in polymer electrolyte melts should include changes in the exponents of the time-related terms.

## Figures and Tables

**Figure 1 ijms-25-04472-f001:**
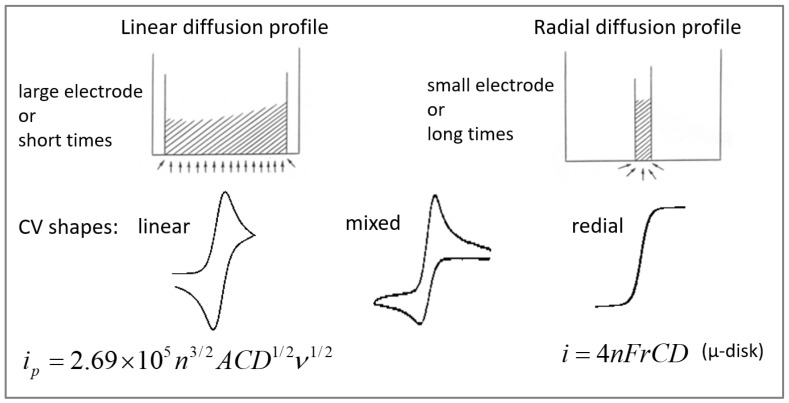
The diffusion profiles, the cyclic voltammograms and the current equations at 25 °C for linear and radial regimes of diffusion.

**Figure 2 ijms-25-04472-f002:**
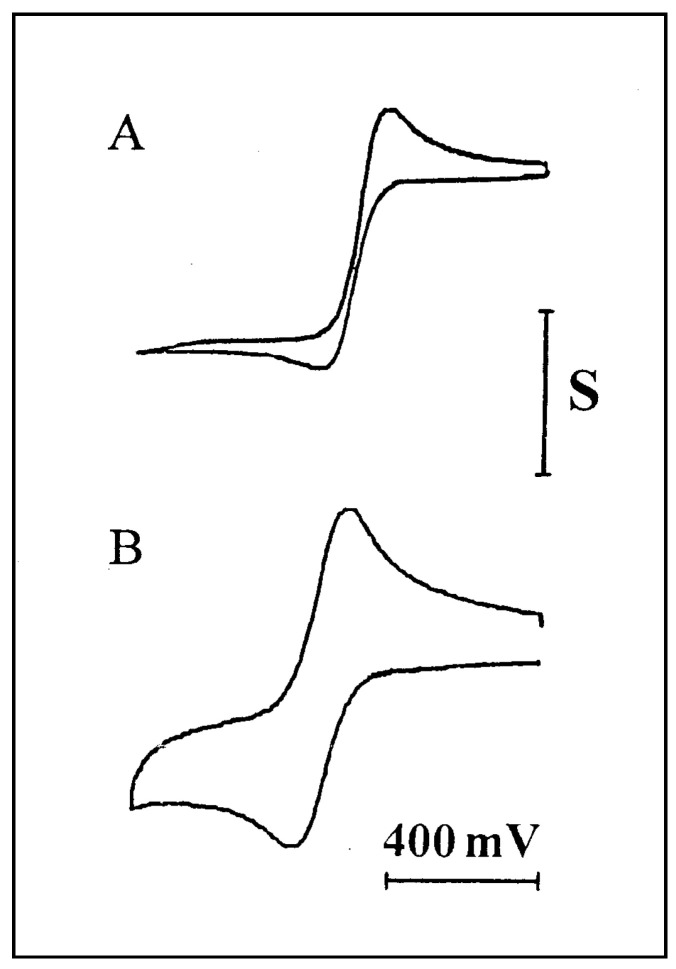
Cyclic voltammograms for 11.3 mM FcCO_2_PEG(350)CH_3_ in PPG 4000/LiSO_4_CF_3_, 1.1 M (O/Li = 16:1). Working electrode: Pt microband, *w* = 2.7 µm. Ref. electrode: parallel Ag bands, *d* = 3 µm. (**A**) *v* = 1 mV/s, S = 2 nA. (**B**) *v* = 200 mV/s. S = 10 nA.

**Figure 3 ijms-25-04472-f003:**
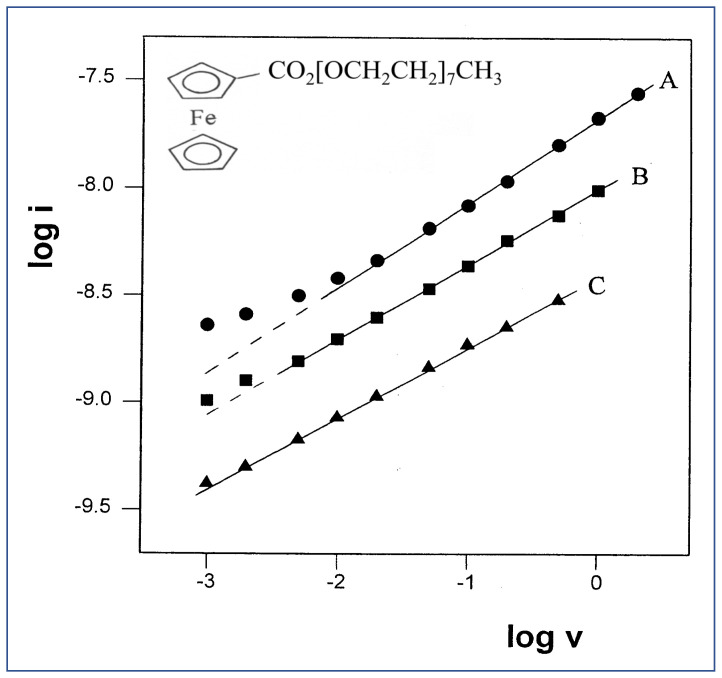
Plot of log i vs. log v for three sets of CV experiments of 11.3 mM FcCO_2_PEG(350)CH_3_ in PPG 4000/LiSO_4_CF_3_, 1.1 M (O/Li = 16:1), at various temperatures. (●) T = 22.8 °C, slope = 0.40. (■) T = 12.6 °C, slope = 0.35. (▲) T = 2.6 °C, slope = 0.33.

**Figure 4 ijms-25-04472-f004:**
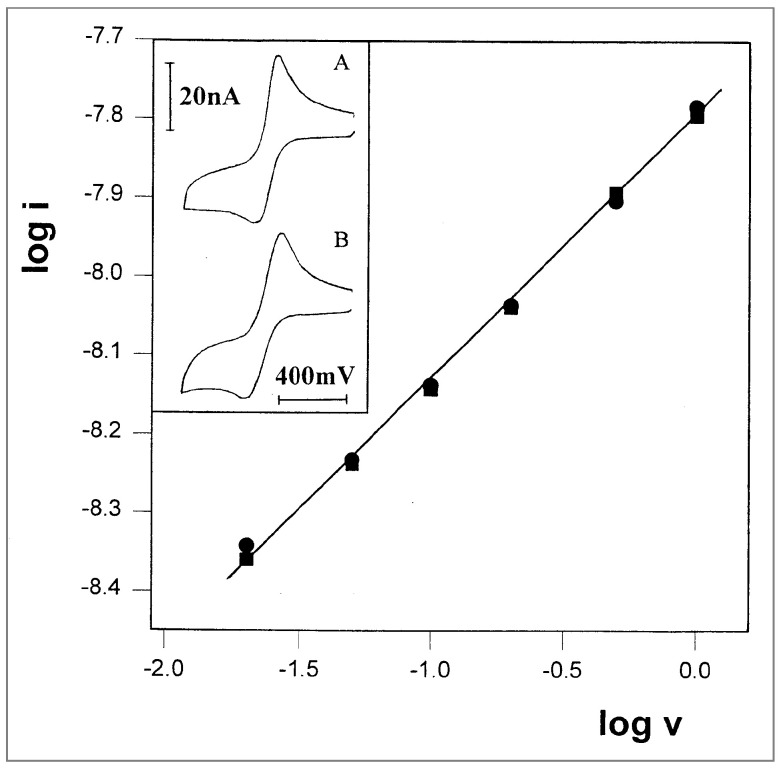
Plot of log i vs. log v for two sets of CV of 11.3 mM FcCO_2_PEG(350)CH_3_ in PPG 4000/LiSO_4_CF_3_, 1.1 M, performed with (■) and without (●) applied positive feedback at room temperature. Slope = 0.33. Inset: representative voltammograms, recorded at 500 mV/s. (A) With applied positive feedback, Δ*Ep* = 156 mV. (B) No applied positive feedback, Δ*Ep* = 218 mV.

**Figure 5 ijms-25-04472-f005:**
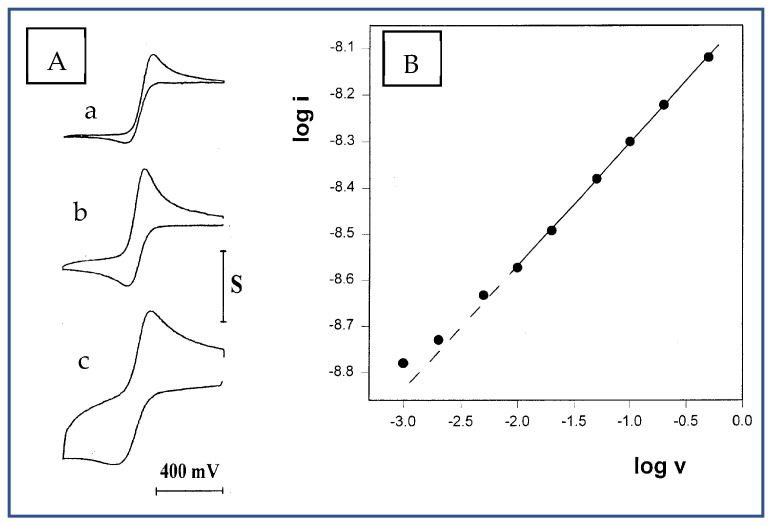
(**A**) Three of the cyclic voltammograms of 10 mM FcCO_2_PEG(350)CH_3_ out of a series recorded in PPG10,000/LiSO_4_CF_3_, 1.1 M, at T = 23 °C. (a) 1 mV/s, S = 2 nA. (b) 20 mV/s, S = 4 nA. (c) 500 mV/s, **S** = 8 nA. (**B**) The corresponding plot of log i vs. log v. Slope = 0.26.

**Figure 6 ijms-25-04472-f006:**
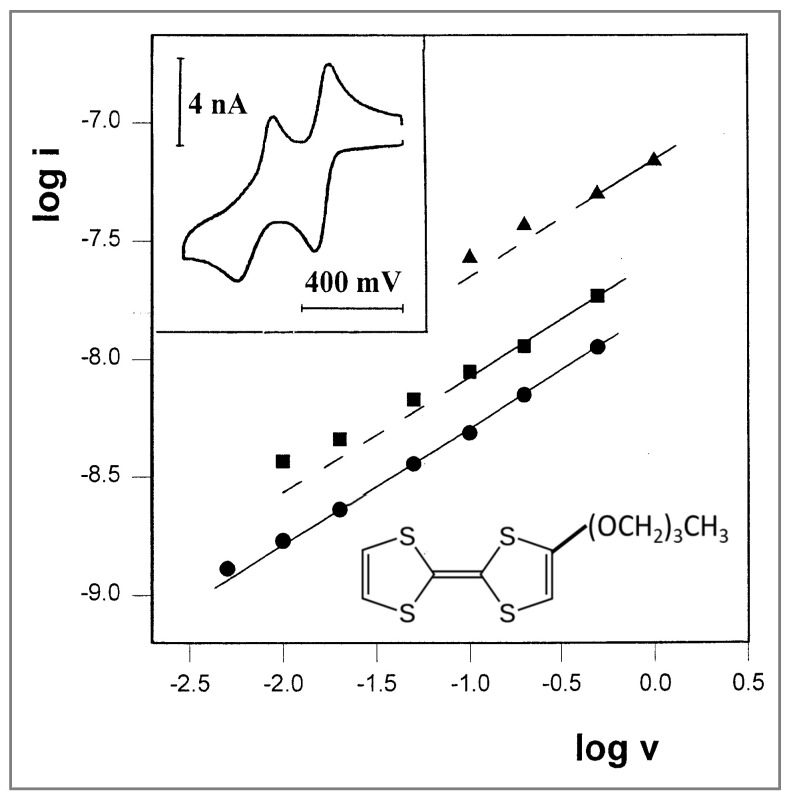
Plot of log i vs. log v for three sets of cyclic voltammograms of 12 mM TTF-(OCH_2_)_3_CH_3_ in propylene glycol/0.1 M LiClO_4_. (●) T = −23.7 °C. (■) T = −12.0 °C. (▲) T = 1.0 °C. All lines are drawn with slopes of 0.5. Inset: A representative voltammogram recorded at a scan rate of 20 mV/s.

**Figure 7 ijms-25-04472-f007:**
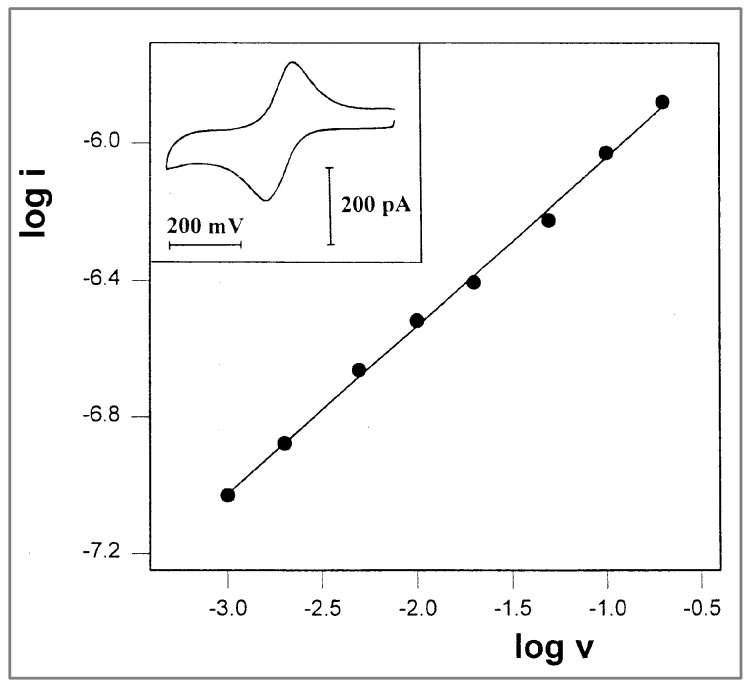
Plot of log i vs. log v for the oxidation peak currents of 10 mM FcCO_2_PEG(350)CH_3_ in glycerol/LiSO_3_CF_3_, 1 M, at T = −27 °C. Slope = 0.49. Inset: A representative voltammogram recorded at a scan rate of 1 mV/s.

**Figure 8 ijms-25-04472-f008:**
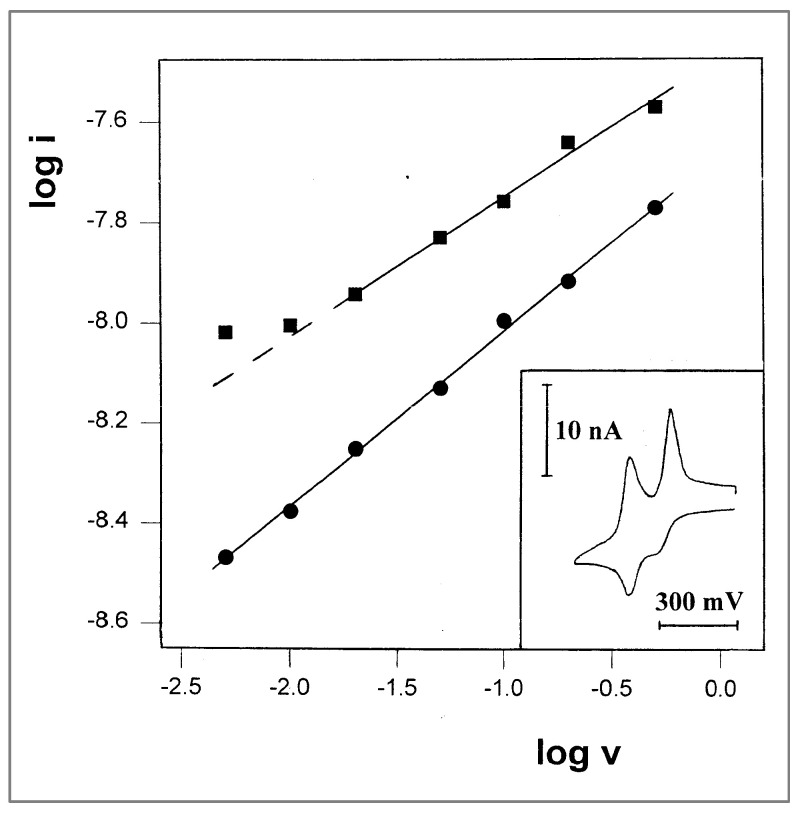
Plot of log i vs. log v for the first oxidation peaks (●) and the first reduction peaks on the reverse scan (■) in the CV of 12 mM TTF-(OCH_2_)_3_CH_3_ in propylene glycol/0.1 M LiClO_4_, recorded at room temperature. Inset: A representative voltammogram recorded at a potential scan rate of 10 mV/s.

**Figure 9 ijms-25-04472-f009:**
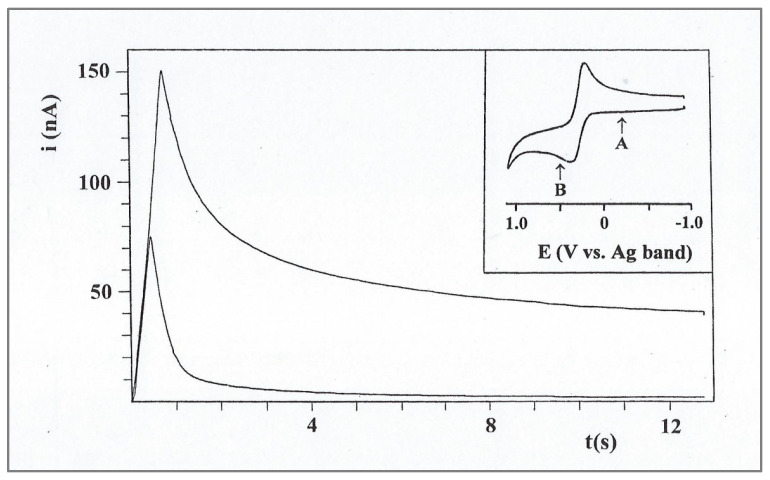
Chronoamperometry of 11.3 mM FcCO_2_PEG(350)CH_3_ in PPG 4000/LiSO_4_CF_3_, 1.1 M, recorded at room temperature. Inset: CV in the same system, showing the potential steps for the background curve (A) and for FcCO_2_PEG(350)CH_3_ (B).

**Figure 10 ijms-25-04472-f010:**
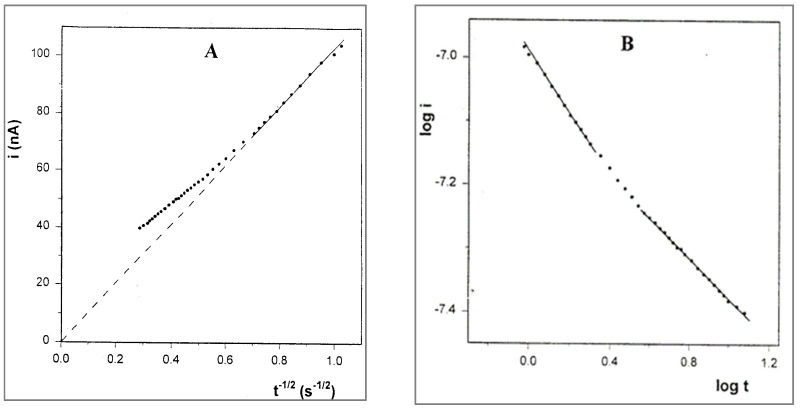
(**A**) Cottrell plot for the chronoamperogram in Figure 8. (**B**) Plot of log i vs. log t corresponding to the same chronoamperogram.

**Figure 11 ijms-25-04472-f011:**
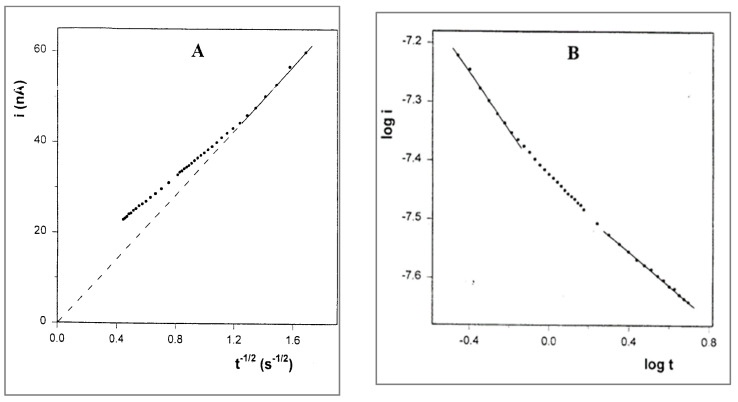
(**A**) Cottrell plot for the chronoamperogram of 10 mM FcCO_2_PEG(350)CH_3_ in PPG10,000/LiSO_4_CF_3_, 1.1 M, at room temperature. (**B**) Plot of log i vs. log t corresponding to the same chronoamperogram.

**Figure 12 ijms-25-04472-f012:**
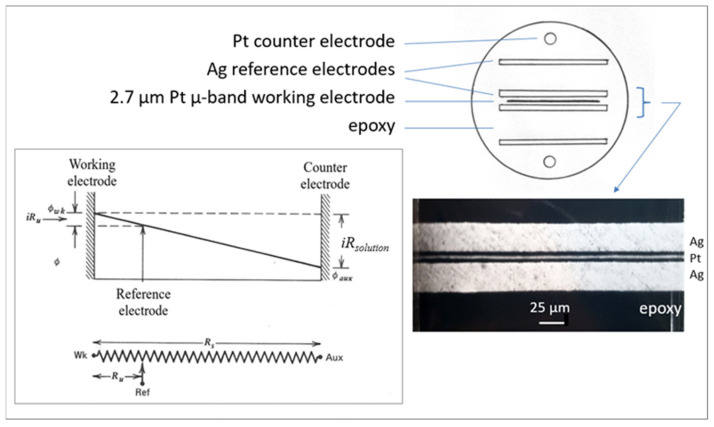
Schematic presentation of the surface of the combined electrodes assembly. The SEM image shows a section of the Au working electrode with two nearby parallel Ag reference electrodes. Inset: The effect of the distance between the reference and the working electrode on *iR*_solution_, which is prominent in resistive media [17].

**Figure 13 ijms-25-04472-f013:**
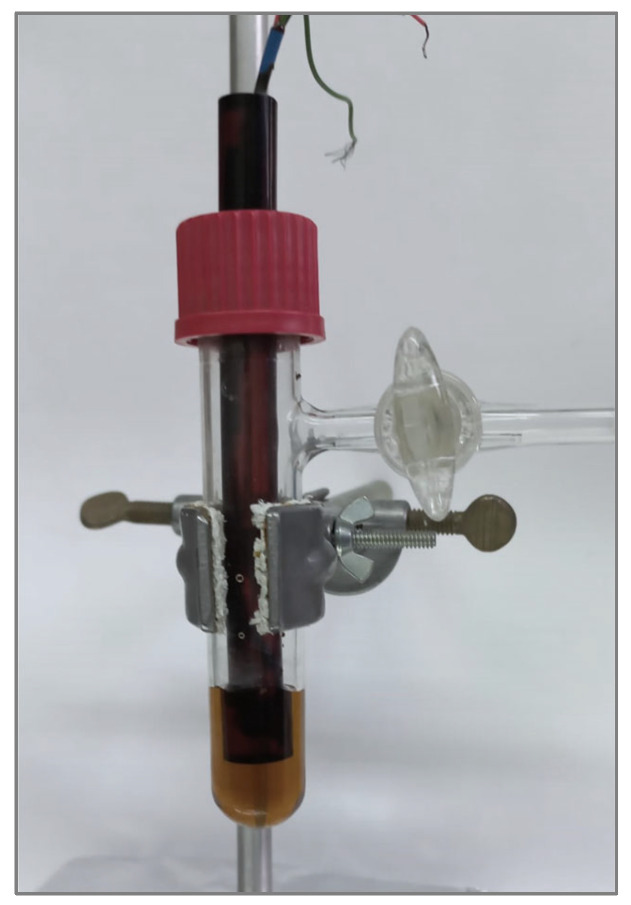
The electrochemical cell: the combined electrode rod is embedded in the polymer electrolyte solution. Removing the plasticizer (ethanol) was carried out by dipping the cell in an oil bath under vacuum. Cooling the solution was accomplished by dipping the cell in a cooling bath mixture composed of dry ice and organic solvent.

## Data Availability

Data contained within the article.

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
