# Peer review of "Electrochemical Characterization of Diffusion in Polymeric vs. Monomeric Solvents"

_ijms, 2024, doi:10.3390/ijms25084472_

Round 1
Reviewer 1 Report
Comments and Suggestions for Authors
The main problem of presented paper is in interpretation of obtained data and in experimental conditions. The polymer electrolytes are the topic of many studies. For interpretation on molecular level necessary to use the dates of the NMR studies which described the ion mobilities on molecular level or segmental for polymers. Most these studies shown that Fick equation adequate described mobility of Li ions. The author should rewrite the paper with using the data of NMR and dielectric studies.
Comments on the Quality of English Language
Probably, need a minor corrections
Author Response
Reply: NMR was indeed used to measure diffusion coefficients in polymer electrolytes. However, this is not the issue in this work. The mobility of the Li ions is caused by migration, which is mass-transfer of charged particles due to the applied voltage between the working electrode to the counter electrode. The role of the ionic mobility is to provide electric conductivity to the system, but it is not part of the electrochemical reaction that is studied here. The electroactive species in this work are FcCO2PEG(350)CH3 and TTF-(OCH2)CH3 that diffuse due to concentration gradients after being oxidized or reduced at the working electrode. The original goal of the experiments was to test the microband electrodes under conditions of linear diffusion in viscous polymeric media. Since we did not get the expected results, it was understood that the diffusional behavior and characteristics in monomeric solvents (free Fickian) is different than in polymer electrolytes (complex diffusion). This is the main issue of this work, and it is now explained more clearly.
Reviewer 2 Report
Comments and Suggestions for Authors
The manuscript deals with a very interesting study of electrochemical characterization of diffusion in polymeric and monomeric solvents.
The author should address the following issues:
1) The introduction should adequately present the current state-of-the-art and afterwards clearly point out the novel aspects and significant contributions of the manuscript. For example, is the any other way to study the diffusion in polymeric and monomeric solutions that does not use electrochemistry? What is the advatage of using an electrochemistry approach?
2) In line 88, page 3, the author must adequately reference the experimental procedure used to synthezise the chemicals.
3) Regarding the used chemicals, the author should provide CAS-Number, degree of purity, supplier, and also mention if it was used as received or an extre purifying step was used.
4) The experiments were conducted in triplicates?
5) The manuscript completely lacks in statistical analysis. For example, what is the uncertainty of the calculated slopes of the lines in Figure 2 and also in the other figures?
6) In page 12, line 337, the author mention: In our polymeric solution the plasticizer was removed,. Firstly, this information should be provided in the section Materials and Methods. Secondly, how was it removed? What procedure was used? How can the author be certain the plasticizer was completely removed? Is there any analysis?
Author Response
1) The introduction should adequately present the current state-of-the-art and afterwards clearly point out the novel aspects and significant contributions of the manuscript. For example, is the any other way to study the diffusion in polymeric and monomeric solutions that does not use electrochemistry? What is the advatage of using an electrochemistry approach?
Reply: The original goal of this work was not studying the diffusion in polymer electrolytes by electrochemistry, but rather using polymer electrolyte to test our microband electrodes in such viscous media, in order to obtain conditions of observing linear diffusion profile to the new type of microband electrode. This is said in the first sentence of the abstract and better explained now in the Introduction. We did not use electrochemistry because of its advantages (and it is advantageous in many way) but basically, we intended to test the characteristics of this special type of electrode, and got some insights into the nature of diffusion in various media.
2) In line 88, page 3, the author must adequately reference the experimental procedure used to synthesize the chemicals.
Reply: This point was clarified as follows: “Ferrocene-labeled monomethyl poly(ethylene glycol) [FcCO2PEG(350)CH3] [16] and tetrathiafulvalene trimethoxy methyl [TTF-(OCH2)CH3] [17] were synthesized according to procedures described in these references.”
3) Regarding the used chemicals, the author should provide CAS-Number, degree of purity, supplier, and also mention if it was used as received or an extre purifying step was used.
Reply: Corrected.
4) The experiments were conducted in triplicates?
Reply: In general, the unexpected results repeated themselves in all the experiments. In particular, when doing a series of cyclic voltammetry measurements, there is no need to repeat each voltammogram three times, especially when the results are well aligned on a straight line in the i vs. ν1/2 or log i vs. log ν. Small variations in i would make no difference and large variations cannot occur in such cases.
5) The manuscript completely lacks in statistical analysis. For example, what is the uncertainty of the calculated slopes of the lines in Figure 2 and also in the other figures?
Reply: Statistical analysis is not required in such voltammetric experiments. As mentioned above, the results show full alignments with the straight lines or the consistent (not random!) deviations that are explained. Small variations can be only in the measured currents, which were measured accurately. Repeating each experiments several times can only increase the variations in the current because the temperatures may not stay exactly the same.
6) In page 12, line 337, the author mention: In our polymeric solution the plasticizer was removed. Firstly, this information should be provided in the section Materials and Methods. Secondly, how was it removed? What procedure was used? How can the author be certain the plasticizer was completely removed? Is there any analysis?
Reply: The removal of the plasticized is mentioned in the Experimental section. The plasticizer was a small amount of ethanol, which was added to facilitate the mixing of the solutions components. It was removed by heating the solution at 70 oC under vacuum for many hours. This should be sufficient, because the boiling temperature of ethanol is 78.4 oC under atmospheric pressure. Analysis was not possible in this system and was not needed either, because the final conclusion was that the structure of the medium molecules and not the viscosity was the reason for the different diffusion modes.
Reviewer 3 Report
Comments and Suggestions for Authors
Dear Uthors
You have decided to face a very complex system with certain success. In general the use of this type of polymeric suspension in ionic media generates a dynamic mechanism of aggregation that probably deviates the fickian diffusion towards an anomalous diffusion.
The changes on molecular weigth just increase these type of phenomena, therefore the results are a little bit expected.
In general, you have to change the format of the chemical expressions used with the subindexes well formatted. It is required to incorporate a figure with a schematic description of the system because it is difficult for unfamiliar readers with these complex phenomena.
My concrns and recommendations are highligthed in the attached file.
The format for bibliographic references should be changed from () to [], and equations should follow the contrary sens.
Page 3, when describing the instruments and reageants, please consider to add city and country. Aldrich is now Merck
page 5, line 147 there are three sets of temperature instead two as you mention
The use of temperature as t, is contrary to the rest of parameters described in the article, please consider to change in the legend of some figures (fig 2)
Many thanks

Author Response
You have decided to face a very complex system with certain success. In general the use of this type of polymeric suspension in ionic media generates a dynamic mechanism of aggregation that probably deviates the fickian diffusion towards an anomalous diffusion.
The changes on molecular weight just increase these types of phenomena, therefore the results are a little bit expected.
1) In general, you have to change the format of the chemical expressions used with the subindexes well formatted.
This remark is unclear to me.
2) It is required to incorporate a figure with a schematic description of the system because it is difficult for unfamiliar readers with these complex phenomena.
Reply: Two figures were added to in the Introduction for a better description of the experimental system.
3) My concerns and recommendations are highlighted in the attached file.
The format for bibliographic references should be changed from () to [], and equations should follow the contrary sens.
Reply: Corrected.
4) Page 3, when describing the instruments and reageants, please consider to add city and country. Aldrich is now Merck.
Reply: Corrected to Merck. Cities and countries are unknown.
5) page 5, line 147 there are three sets of temperature instead two as you mention
Reply: Corrected.
6) The use of temperature as t, is contrary to the rest of parameters described in the article, please consider to change in the legend of some figures (fig 2).
Reply: Corrected.
Round 2
Reviewer 1 Report
Comments and Suggestions for Authors
It is a second round of review the paper of Ze’ev Porat
I start my comments from Conclusions
1. Diffusion of molecules in polymer electrolyte melt is non-Fickian because of various perturbations to free random motion derived from concentrations gradients.
The diffusion in all medium govern by concentration gradient.
Therefore, such motivation can not be used for explanation of observed effect.
In my first report I suggest to author to consider the data of NMR study. The number of these studies published, for example, in Macromolecules shown that diffusion of Li ions in polyethylene oxide matrix good described by Fick law.
I suggest to author to consider the possible oxidation of ferrocene containing molecules in conditions of your experiments.
Comments on the Quality of English Language
The English needs in editing
Author Response
I wish to reply thoroughly to the reviewer's remarks, as follows:
1. Remark: Diffusion of molecules in polymer electrolyte melt is non-Fickian because of various perturbations to free random motion derived from concentrations gradients.
Reply: The perturbations to free random diffusion are not derived from concentration gradients, but rather from the complex structure of the polymer-electrolyte melt. The fact that not all diffusion paths are available makes it non-Fickian diffusion.
2. Remark: The diffusion in all medium govern by concentration gradient.
Therefore, such motivation can not be used for explanation of observed effect.
Reply: Diffusion in all media is always govern by concentration gradient, but it is not always Fickian, as mentioned above. The concentration gradient was not the motivation to this work, This work started as an attampt to observe linear diffusion at micrometric microband electrodes in polymer electrilytes as viscous media. Once we failed to observe that, we understood that the diffusion in such media is non-Fickian, unlike monomeric solvents of similar viscosities.
3. Remark: In my first report I suggest to author to consider the data of NMR study. The number of these studies published, for example, in Macromolecules shown that diffusion of Li ions in polyethylene oxide matrix good described by Fick law.
Reply: There are three modes of mass-transfer in electrochemical systems: convection, migration and diffusion. Convection means stirring the solution which enhances mass-transfer very much. Migration is caused by applying electric field: Anions then move towards anode and cations move to the cathode. This motion of the supporting electrilyte provides the electrical conductivity to the system. In our case, the ions of the lithium triflate salt move by migration, not diffusion. In voltammetric experiments, applying a certain voltage at the working electrode causes the electroactive species to be oxidized or reduced so that a concentration gradient is formed between the electrode and the bulk of the solution. This causes diffusion of that species, in Fickian or non-Fickian mode. Our entire system is electrochemical and cannot be performed in an NMR device. How can a ferrocene derivative be oxidized within an NMR tube without applyting a potential scan or step? What kind of information can NMR provide on the ferrocene molecures in the two different oxidation states?
3. Remark: I suggest to author to consider the possible oxidation of ferrocene containing molecules in conditions of your experiments.
Reply: Most of the work describes the oxidation of Ferrocene (Fc) derivetive (FcCO2PEG(350)CH3), See Fig. 5.
As for the English language, the manuscript was revised by a native English-speaking scientist.
Reviewer 2 Report
Comments and Suggestions for Authors
The revised version of the manuscript can be accepted .
Author Response
I thank the reviewer for his remarls and for approving the article.
Reviewer 3 Report
Comments and Suggestions for Authors
Dear Authors,
Many thanks for the efforts in revising and taking into account our recommendations. Now, the article is better for sightseeing and it would be useful for readers
Author Response
I thank the reviewer for his important comments and for approving the article.
Round 3
Reviewer 1 Report
Comments and Suggestions for Authors
The author working with your manuscript without any respect to reviewers and editor. The corrections in the first version are superimposed on the corrections in the second, so it is difficult to understand what has been done. However, the Editor must decide to publish this paper in this form or not.
Comments on the Quality of English Language
The English is appropriate for publication after minor revision